# Incidence and prognostic value of pulmonary embolism in COVID-19: A systematic review and meta-analysis

**Xiaowei Gong, Boyun Yuan, Yadong Yuan** [ORCID] *

Department of Respiratory and Critical Care Medicine, The Second Hospital of Hebei Medical University, Shijiazhuang, China

* yuanyd1108@163.com

## Abstract

**Data Availability Statement:** All relevant data are within the paper and its Supporting Information files.

**Funding:** This study was supported by Hebei Province Science and Technology Support

### Background

Pulmonary embolisms are frequently and prognostically in individuals infected by coronavirus disease 2019 (COVID-19); the incidence of pulmonary embolisms is varied across numerous studies. This study aimed to assess the pooled incidence of pulmonary embolic events and the prognostic value of such events in intensive care unit (ICU) admissions of patients with COVID-19.

### Methods

The Cochrane Library, PubMed, and EmBase were systematically searched for eligible studies published on or before October 20, 2021. The pooled incidence of pulmonary embolism was calculated using the random-effects model. Moreover, the prognostic value was assessed by measuring the sensitivity, specificity, positive and negative likelihood ratio (PLR and NLR), diagnostic odds ratio (DOR), and the area under the receiver operating characteristic curve (AUC).

### Results

Thirty-six studies involving 10,367 COVID-19 patients were selected for the final meta-analysis. The cumulative incidence of pulmonary embolism in patients with COVID-19 was 21% (95% confidence interval [95%CI]: 18−24%; $P<0.001$), and the incidence of pulmonary embolism in ICU and non-ICU patients was 26% (95%CI: 22−31%; $P<0.001$) and 17% (95%CI: 14−20%; $P<0.001$), respectively. The predictive role of pulmonary embolism in ICU admission was also assessed, and the sensitivity, specificity, PLR, NLR, DOR, and AUC were 0.31 (95%CI: 0.21−0.42), 0.84 (95%CI: 0.75−0.90), 1.88 (95%CI: 1.45−2.45), 0.83 (95%CI: 0.75−0.91), 2.25 (95%CI: 1.64−3.08), and 0.61 (95%CI: 0.57−0.65), respectively.

Program (20277706D) and Scientific Research Fund project of Hebei Provincial Health and Family Planning Commission (20170091). The funders had no role in the study design, data collection and analysis, decision to publish, or preparation of the manuscript.

**Competing interests:** The authors declare that they have no known competing financial interests or personal relationships that could have appeared to influence the work reported in this paper.

## Conclusion

This study found that the incidence of pulmonary embolism was relatively high in COVID-19 patients, and the incidence of pulmonary embolism in ICU patients was higher than that in non-ICU patients.

## Introduction

The severe acute respiratory syndrome caused by the novel coronavirus (SARS-CoV-2) was first reported as a cluster of cases in December 2019 [1]. The disease grew into the scale of a global pandemic rapidly, with 93,194,922 confirmed cases, and more than 2 million deaths have been reported as of January 2021 [2]. The clinical presentation of the coronavirus disease 2019 (COVID-19) varies, creating challenges in determining the optimal management of patients infected with COVID-19. The primary cause of fatality in patients with COVID-19 was atypical acute respiratory distress syndrome (ARDS) due to the dissociation between well-conserved lung compliance and severe hypoxemia caused by pulmonary vasoregulation disruption and local thrombogenesis [3, 4]. Moreover, coagulopathy was reported in COVID-19 patients presenting with high levels of D-dimer, which causes inflammation (cytokine storm) and coagulation activation, and is associated with an increased risk of mortality [5–7]. However, the incidence of pulmonary embolism in patients in the intensive care unit (ICU) versus the general ward remains uncertain.

Several systematic reviews and meta-analyses have addressed the incidence of pulmonary embolism in patients with COVID-19 [8–12]. Suh et al. identified studies published before June 15, 2020, and found that the pooled incidence of pulmonary embolism was 16.5% and was significantly higher in ICU patients than in the non-ICU patients (24.7% versus 10.5%) [8]. Malas et al. found that the pooled incidence of pulmonary embolism was 13% in non-ICU and 19% in ICU patients [9]. Liao et al. reviewed 19 studies and found the pooled incidence of pulmonary embolism in COVID-19 patients was 15.3% [10]. Roncon et al. identified 23 studies and found that the pooled in-hospital incidence of pulmonary embolism was 23.4% in COVID-19 patients in the ICU and 14.7% in those in the general wards [11]. Lu et al. found that the pooled incidence of pulmonary embolism was 15% but did not find any significant effect on mortality rate when using anticoagulation therapy in hospitalized COVID-19 patients [12]. However, the pooled incidence for pulmonary embolism varied due to the varying severity of the included COVID-19 cases. Moreover, the most recent data were not included, and the results need to be updated. Therefore, this study was conducted to update the pooled incidence of pulmonary embolism in ICU and non-ICU patients infected with COVID-19. Furthermore, the predictive value of pulmonary embolism for ICU admission was also assessed.

## Methods

### Data sources, search strategy, and selection criteria

This systematic review and meta-analysis were performed and reported following the Preferred Reporting Items for Systematic Reviews and Meta-Analysis Statement (PRISMA Checklist) [13]. The eligible studies reported the incidence of pulmonary embolism in patients infected by COVID-19 and included whether the patients were admitted to the ICU. PubMed, EmBase, and the Cochrane Library were systematically searched for eligible studies published on or before October 20, 2021, and the following search terms were applied: ((("COVID-

19"[Supplementary Concept] OR "Coronavirus"[MeSH Terms]) OR "severe acute respiratory syndrome coronavirus 2"[Supplementary Concept]) OR ((((("Coronavirus"[MeSH Terms] OR "Coronavirus"[All Fields]) OR "coronaviruses" [All Fields]) OR (((((((("COVID-19"[All Fields] OR "covid 2019"[All Fields]) OR "severe acute respiratory syndrome coronavirus 2"[Supplementary Concept]) OR "severe acute respiratory syndrome coronavirus 2"[All Fields]) OR "2019 ncov"[All Fields]) OR "sars cov 2"[All Fields]) OR "2019ncov"[All Fields]) OR (("wuhan"[All Fields] AND ("Coronavirus"[MeSH Terms] OR "Coronavirus"[All Fields]))))))) OR (("severe acute respiratory syndrome coronavirus 2"[Supplementary Concept] OR "severe acute respiratory syndrome coronavirus 2"[All Fields]) OR "2019 ncov"[All Fields])) OR (("severe acute respiratory syndrome coronavirus 2"[Supplementary Concept] OR "severe acute respiratory syndrome coronavirus 2"[All Fields]) OR "sars cov 2"[All Fields]))) AND (("embol*"[All Fields] OR "thromb*"[All Fields]) OR ("Embolism"[MeSH Terms] OR "Thrombosis"[MeSH Terms])). The reference lists of the relevant review and original articles were also reviewed manually to identify any new eligible study that met the inclusion criteria.

The literature search and study selection were independently performed by two reviewers, and conflicts between the reviewers were resolved by mutual discussion. The inclusion criteria were as follows: (1) study design: no restrictions placed on study design, including prospective cohort, retrospective cohort, and case series; (2) participants: patients with COVID-19; and (3) exposure and outcomes: pulmonary embolism incidence after admission in ICU or non-ICU patients. Case reports and matched case-control studies were excluded since the incidence of pulmonary embolism was potentially biased.

## Data collection and quality assessment

The data items were independently extracted by two authors and included the names of the first authors of the article, publication year, study design, country, sample size, mean age, male proportion, setting, body mass index (BMI) of the patients, diabetes mellitus status (DM), hypertension status, cardiovascular disease status, and the number of pulmonary embolisms. Subsequently, the two authors independently assessed the quality of each individual study using the Newcastle-Ottawa Scale (NOS), which is based on selection (4 items), comparability (1 item), and outcome (3 items) [14]. Any disagreement between the authors regarding data abstracted and/or quality assessment was settled by mutual discussion until a consensus was reached.

## Statistical analysis

The incidence of pulmonary embolism was based on the occurrence of the event and the total sample size in the cohort. Pooled incidence with 95% confidence intervals (CIs) was calculated using the random-effects model, which considered the underlying variability among the included studies [15, 16]. The sensitivity, specificity, positive likelihood ratio (PLR), negative likelihood ratio (NLR), diagnostic odds ratio (DOR), and area under the receiver operating characteristic curve (AUC) for the role of pulmonary embolism in subsequent ICU admission were calculated, based on the number of ICU admissions in the patient group with a pulmonary embolism and the group without a pulmonary embolism. The pooled analysis was based on the bivariate generalized linear mixed model and random-effects model [15–17]. The heterogeneity across the included studies was assessed using $I^2$ and Q statistics, and significant heterogeneity was defined as $I^2 > 50.0\%$ or $P < 0.10$ [18, 19]. Subgroup analyses were performed for the pooled incidence of pulmonary embolism based on ICU admission or non-admission. Moreover, subgroup analyses were also conducted for the DOR based on the study design, mean age, male proportion, and study quality. Publication biases were assessed by using the

funnel plot and Egger and Begg tests [20, 21]. All reported inspection levels were two-sided, and $P<0.05$ was regarded as statistically significant. All statistical analyses in this study were performed using STATA (version 10.0; Stata Corporation, College Station, TX, USA) and Meta-DiSc (version 1.4; Universidad Complutense, Spain) software.

# Results

## Literature search

The initial electronic search yielded 8,346 articles, with 5,642 remaining after the duplicate articles were removed. Subsequently, 5,483 articles were excluded because of irrelevant topics. The remaining 159 studies were retrieved for further full-text evaluations, and 123 studies were excluded as they did not report pulmonary embolism (n = 47), had a case-control or case report structure (n = 42), and had no ICU admission information included (n = 34). The cited references included in the articles were reviewed; however, no additional eligible studies were found. Thus, the remaining 36 studies were selected for the final meta-analysis (Fig 1) [22–57].

## Study characteristics

The baseline characteristics of the identified and selected studies are shown in Table 1. Of the 36 included studies, six used a prospective design, while the remaining 30 were retrospective studies or case series. All studies were conducted in Europe, and the sample size ranged from 23 to 2,233, with a total of 10,367 patients with COVID-19. Thirty studies reported the incidence of pulmonary embolism in patients with COVID-19 admitted to the ICU, and 25 studies reported the incidence of pulmonary embolism in COVID-19 patients admitted to the general ward. Nine studies were categorized as having a relatively high quality with an NOS of seven stars, while 12 studies were rated as six stars, 11 as five stars, and four as four stars.

## Pooled incidence

After pooling all the included studies, we noted that the pooled incidence of pulmonary embolism in patients with COVID-19 was 21% (95%CI: 18−24%; $P<0.001$; Fig 2), and significant heterogeneity was observed among the included studies ($I^2$ = 95.0%; $P<0.001$). Subgroup analyses revealed that the incidence of pulmonary embolism in patients with COVID-19 admitted to the ICU was 26% (95%CI: 22−31%; $P<0.001$), while that in patients with COVID-19 admitted to the general ward was 17% (95%CI: 14−20%; $P<0.001$).

## Predictive value of pulmonary embolism

Seventeen studies reported the predictive value of pulmonary embolism for ICU admission. The pooled sensitivity and specificity were 0.31 (95%CI: 0.21−0.42) and 0.84 (95%CI: 0.75 −0.90), respectively (Fig 3). There was significant heterogeneity in sensitivity ($I^2$ = 92.9%; $P<0.01$) and specificity ($I^2$ = 96.4%; $P<0.01$). Moreover, the pooled PLR and NLR were 1.88 (95%CI: 1.45−2.45) and 0.83 (95%CI: 0.75−0.91), respectively (Fig 4). We noted potential significant heterogeneity in the PLR ($I^2$ = 36.8%; $P<0.01$) and NLR ($I^2$ = 62.4%; $P<0.01$). Furthermore, the DOR for the predictive value of pulmonary embolism for ICU admission was 2.25 (95%CI: 1.64−3.08; $P<0.001$; Fig 5), and there was significant heterogeneity in the DOR ($I^2$ = 51.7%; $P$ = 0.007). Lastly, the AUC for the role of pulmonary embolism in ICU admission was 0.61 (95%CI: 0.57−0.65; Fig 6).

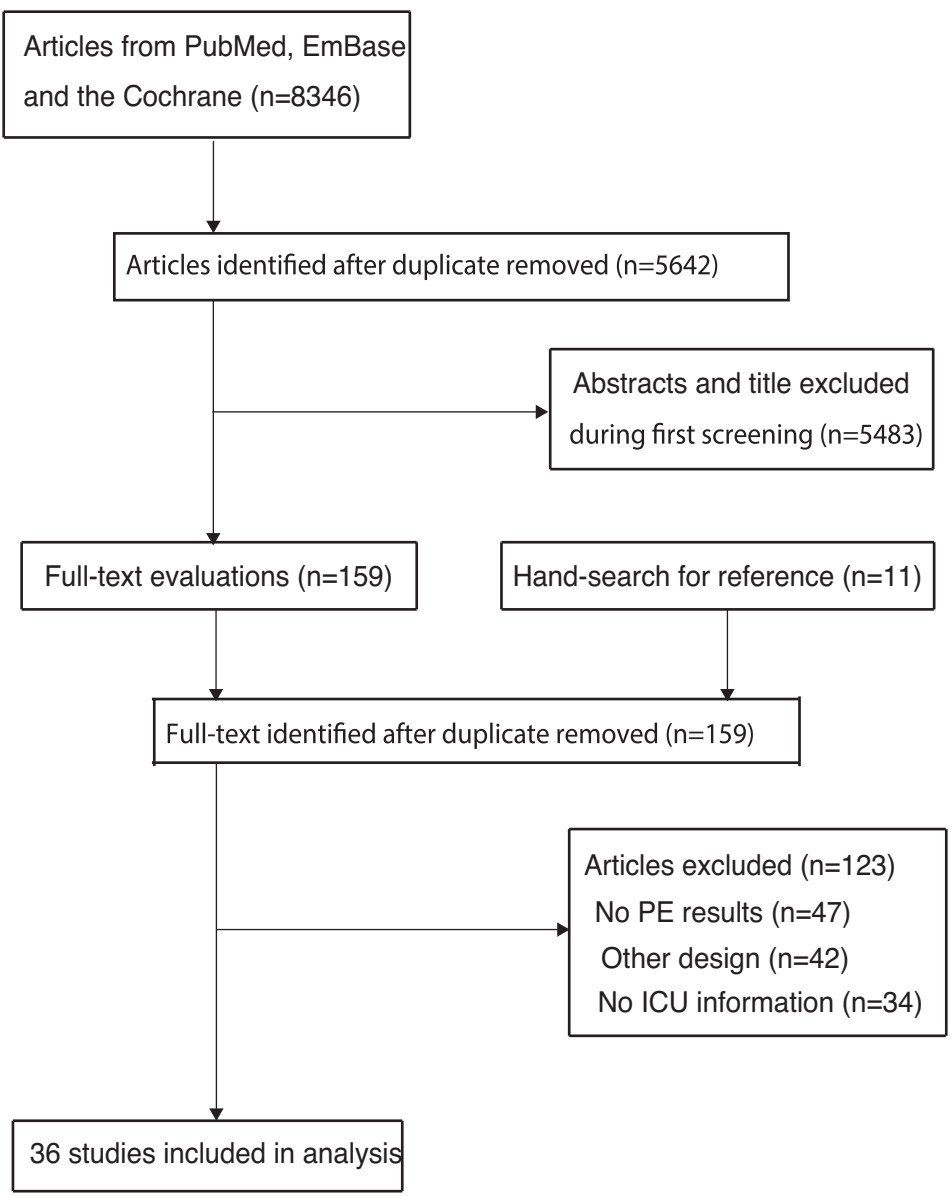

**Fig 1. The flow diagram for the literature search and study selection.**

## Subgroup analysis for DOR

The results of subgroup analyses performed for the DOR are shown in Table 2. Although the DOR of pulmonary embolism for ICU admission was statistically significant in most sub-groups, this predictive value was not statistically significant when prospective studies were pooled. Moreover, we noted that the DOR of pulmonary embolism for ICU admission could be affected by the study design ($P = 0.046$) and male proportion ($P = 0.030$).

## Publication bias

The publication biases for the incidence of pulmonary embolism and the predictive value of pulmonary embolism for ICU admission are listed in Fig 7. We noted significant publication

**Table 1. The baseline characteristics of included studies and participants.**

| Study | Study design | Country | Sample size | Mean age (years) | Male (%) | Setting | BMI (kg/m$^2$) | DM (%) | Hypertension (%) | CVD (%) | Number of PE | NOS score |
|---|---|---|---|---|---|---|---|---|---|---|---|---|
| Leonard-Lorant 2020 [22] | Retrospective | France | 106 | 63.3 | 66.0 | ICU (48)/non-ICU (58) | 28.4 | NA | NA | NA | 32 | 6 |
| Grillet 2020 [23] | Retrospective | France | 100 | 66.0 | 70.0 | ICU (39)/non-ICU (61) | NA | 20.0 | NA | 39.0 | 23 | 6 |
| Helms 2020 [24] | Prospective | France | 150 | 63.0 | 81.3 | ICU (150) | NA | 20.0 | NA | 48.0 | 25 | 7 |
| Hékimian 2020 [25] | Retrospective | France | 51 | NA | NA | ICU (8) | NA | NA | NA | NA | 8 | 5 |
| Fraissé 2020 [26] | Retrospective | France | 92 | 61.0 | 79.0 | ICU (92) | 30.0 | 38.0 | 64.0 | 10.0 | 19 | 5 |
| Van Dam 2020 [27] | Retrospective | Netherlands | 23 | 63.0 | 70.0 | Non-ICU (23) | NA | NA | NA | NA | 4 | 4 |
| Gervaise 2020 [28] | Retrospective | France | 72 | 62.0 | 75.0 | Non-ICU (72) | 26.7 | NA | NA | NA | 13 | 6 |
| Soumagne 2020 [29] | Retrospective | France | 375 | 63.5 | 76.8 | ICU (375) | 29.8 | 26.4 | 57.6 | 9.6 | 55 | 7 |
| Longchamp 2020 [30] | Retrospective | Switzerland | 25 | 68.0 | 64.0 | ICU (25) | 27.5 | 4.0 | 40.0 | 12.0 | 5 | 5 |
| Klok 2020 [31] | Retrospective | Netherlands | 184 | 64.0 | 76.0 | ICU (184) | NA | NA | NA | NA | 65 | 5 |
| Llitjos 2020 [32] | Retrospective | France | 26 | 68.0 | 76.9 | ICU (26) | NA | NA | 85.0 | NA | 6 | 4 |
| Bompard 2020 [33] | Retrospective | France | 135 | 64.0 | 70.0 | ICU (24)/non-ICU (111) | NA | NA | NA | NA | 32 | 6 |
| Artifoni 2020 [34] | Retrospective | France | 71 | 64.0 | 60.6 | Non-ICU (71) | 27.3 | 20.0 | 41.0 | NA | 7 | 5 |
| Thomas 2020 [35] | Retrospective | UK | 63 | NA | 69.0 | ICU (63) | NA | NA | NA | NA | 5 | 5 |
| Lodigiani 2020 [36] | Retrospective | Italy | 388 | 66.0 | 68.0 | ICU (61)/non-ICU (327) | NA | 22.7 | 47.2 | 13.9 | 10 | 6 |
| Whyte 2020 [37] | Retrospective | UK | 214 | 61.5 | 60.2 | ICU (78)/non-ICU (136) | NA | NA | NA | NA | 80 | 7 |
| Fauvel 2020 [38] | Retrospective | France | 1,240 | 64.0 | 58.1 | ICU (185)/non-ICU (1,055) | 28.1 | 21.7 | 45.4 | 10.7 | 103 | 7 |
| Poissy 2020 [39] | Retrospective | France | 107 | NA | NA | ICU (107) | NA | NA | NA | NA | 22 | 6 |
| Marone 2020 [40] | Retrospective | Italy | 41 | 65.0 | 70.7 | ICU (15)/non-ICU (26) | NA | NA | NA | NA | 24 | 4 |
| Stoneham 2020 [41] | Retrospective | UK | 274 | NA | NA | Non-ICU (274) | NA | NA | NA | NA | 16 | 5 |
| Mestre-Gómez 2020 [42] | Retrospective | Spain | 91 | 64.7 | 68.1 | Non-ICU (91) | 29.5 | 17.6 | 51.6 | NA | 29 | 4 |
| Middeldorp 2020 [43] | Retrospective | Netherlands | 198 | 61.0 | 66.0 | ICU (75)/non-ICU (123) | 27.0 | NA | NA | NA | 13 | 7 |
| Galeano-Valle 2020 [44] | Prospective | Spain | 785 | NA | NA | Non-ICU (785) | NA | NA | NA | NA | 15 | 5 |
| Benito 2020 [45] | Prospective | Spain | 76 | 62.5 | 67.1 | ICU (25)/non-ICU (51) | 27.3 | 15.8 | 44.7 | NA | 32 | 6 |
| Alonso-Fernandez 2020 [46] | Prospective | Spain | 30 | 64.5 | 63.3 | ICU (11)/non-ICU (19) | 28.2 | NA | NA | 40.0 | 15 | 7 |
| Contou 2020 [47] | Retrospective | France | 92 | 61.0 | 79.0 | ICU (92) | NA | 38.0 | 64.0 | 10.0 | 16 | 6 |
| Freund 2020 [48] | Retrospective | France | 974 | 61.0 | 48.0 | Non-ICU (974) | NA | NA | NA | NA | 146 | 5 |
| Scudiero 2020 [49] | Retrospective | Italy | 224 | 69.0 | 62.0 | ICU (73)/non-ICU (151) | NA | 28.0 | 61.0 | 16.0 | 32 | 7 |
| Brüggemann 2020 [50] | Retrospective | Netherlands | 60 | 68.0 | 70.0 | ICU (13)/non-ICU (47) | NA | NA | NA | NA | 24 | 5 |
| Cerda 2020 [51] | Retrospective | Spain | 92 | 68.0 | 73.9 | ICU (26)/non-ICU (66) | 29.2 | 29.3 | 56.5 | 12.0 | 29 | 6 |
| Schmidt 2021 [52] | Prospective | France | 2,233 | 63.0 | 74.0 | ICU (2,233) | 28.0 | NA | NA | NA | 207 | 6 |
| García-Ortega 2021 [53] | Prospective | Spain | 73 | 65.4 | 71.0 | ICU (25)/non-ICU (48) | 29.3 | 18.0 | 50.0 | 10.0 | 26 | 7 |
| Valle 2021 [54] | Retrospective | Italy | 114 | 61.0 | 73.7 | ICU (28)/non-ICU (86) | NA | 14.9 | 35.9 | 44.7 | 65 | 6 |
| Nordberg 2021 [55] | Retrospective | Sweden | 1,162 | 64.0 | 78.0 | ICU (252)/non-ICU (910) | 27.0 | NA | NA | NA | 41 | 6 |
| Badr 2021 [56] | Retrospective | Saudi Arabia | 159 | 52.8 | 69.8 | ICU (119)/non-ICU (40) | 27.8 | 42.1 | 37.1 | 17.0 | 51 | 7 |
| Filippi 2021 [57] | Retrospective | Italy | 267 | 69.9 | 64.4 | ICU (55)/non-ICU (212) | NA | NA | NA | NA | 50 | 5 |

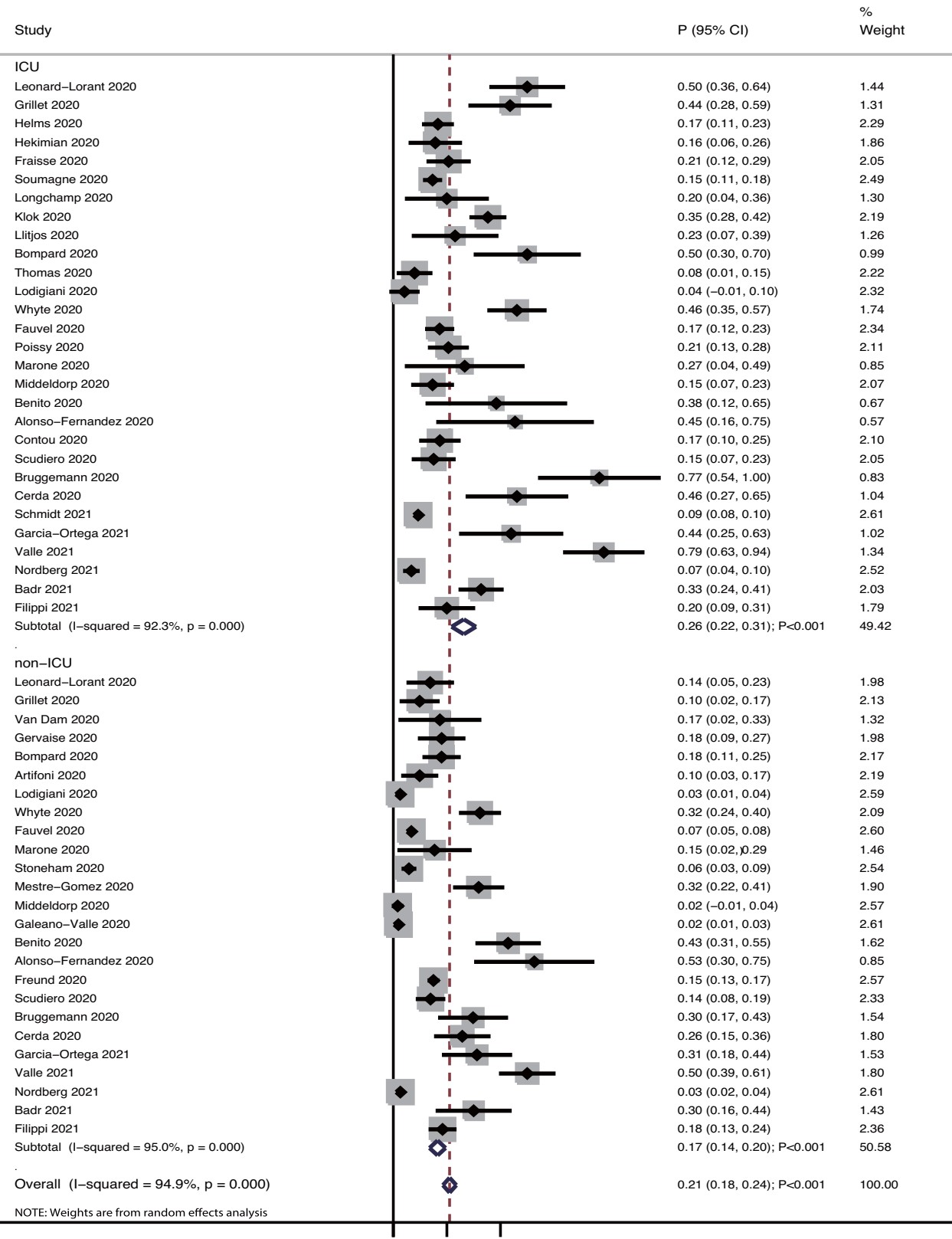

**Fig 2. The pooled incidence of pulmonary embolism in COVID-19 patients.**

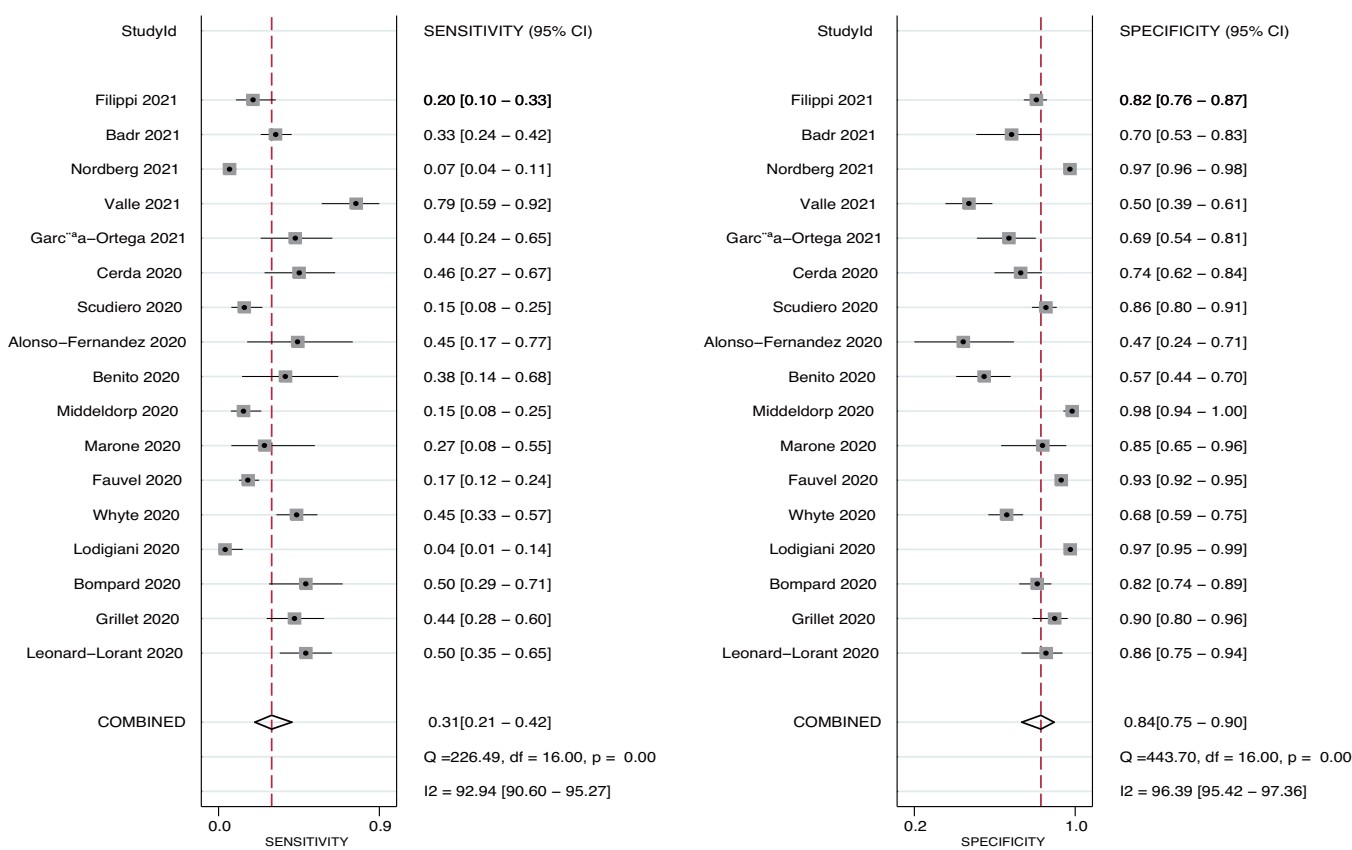

**Fig 3. Pooled sensitivity and specificity for pulmonary embolism on subsequent ICU admission.**

bias for the incidence of pulmonary embolism (*P*-value for Egger: <0.001; *P*-value for Begg: <0.001); after adjusting for potential publication bias, the pooled incidence changed to 8% (95%CI: 6%-11%; *P*<0.001). Moreover, there was no significant publication bias for the predictive value of pulmonary embolism for ICU admission (*P* = 0.61).

## Discussion

The incidence of pulmonary embolism in patients with COVID-19 varies, and whether pulmonary embolism could predict ICU admission remains inconclusive. A total of 36 studies involving 10,367 COVID-19 patients was selected in this quantitative analysis, and the characteristics of these studies and patients were broad. This study found that the pooled incidence of pulmonary embolism was 21% (95%CI: 18–24%), and the incidence of pulmonary embolism in ICU patients was higher than that in non-ICU patients with COVID-19. Moreover, we noted a mild predictive value of pulmonary embolism for subsequent ICU admission in patients infected by COVID-19. Lastly, the predictive value of pulmonary embolism for subsequent ICU admission could be affected by the study design and male proportion.

Several systematic reviews and meta-analyses have already illustrated the pooled incidence of pulmonary embolism in COVID-19 patients. However, these previous reviews combined the pooled incidence of pulmonary embolism according to ICU status, while the predictive role of pulmonary embolism for ICU admission was not illustrated [8–12]. Moreover, while the initial phases of COVID-19 yielded numerous new studies, recent meta-analyses and the pooled incidence of pulmonary embolism in this patient group should be updated. Therefore,

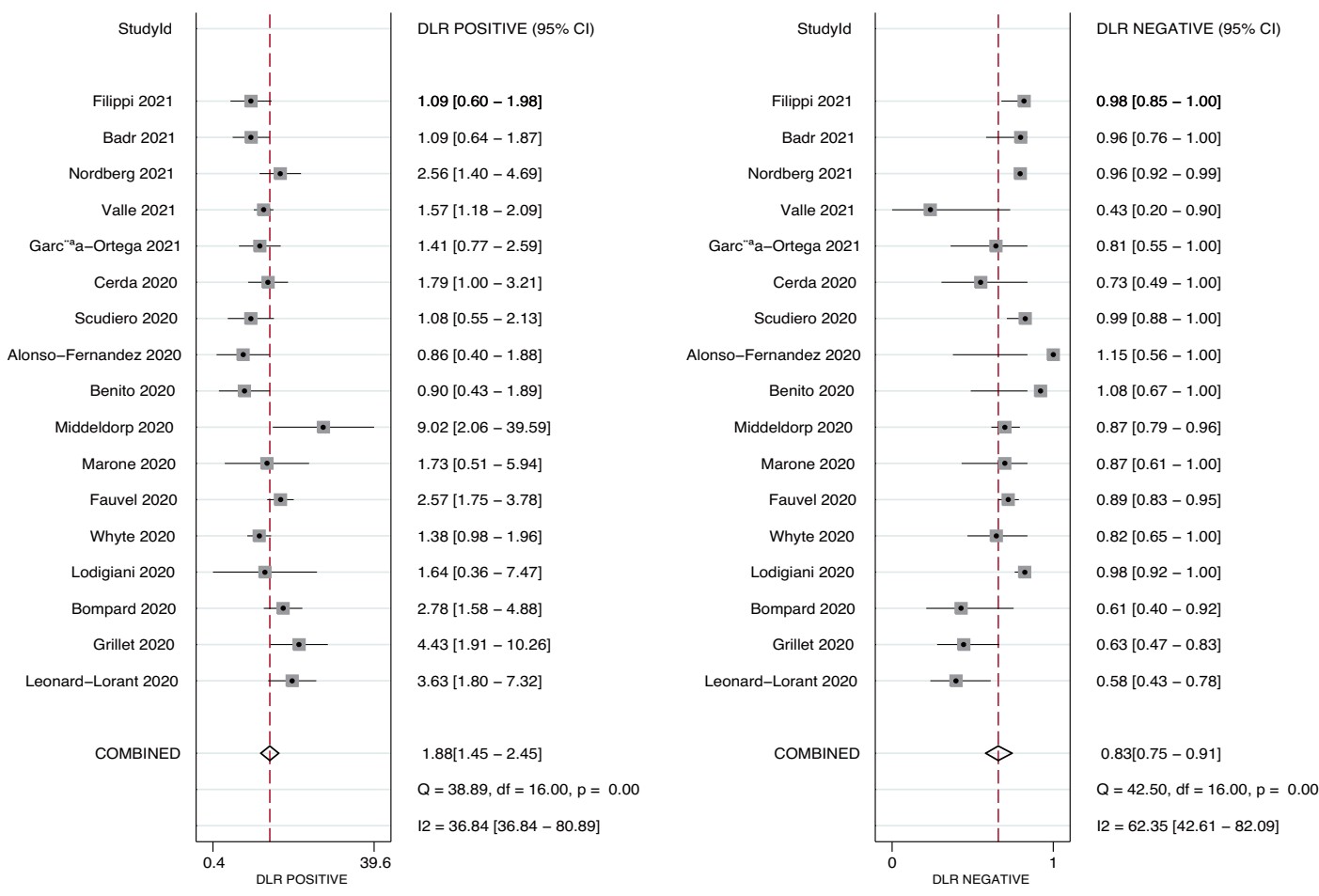

**Fig 4. Pooled PLR and NLR for pulmonary embolism on subsequent ICU admission.**

this systematic review and meta-analysis were performed to update the pooled incidence of pulmonary embolism in patients with COVID-19 patients and assess the predictive role of pulmonary embolism for ICU admission in patients with COVID-19.

This study found that the pooled incidence of pulmonary embolism for COVID-19 patients was 21%, and the incidence in each study ranged from 2–79%. Sensitivity analysis found that the pooled incidence of pulmonary embolism in COVID-19 patients ranged from 20.0–21.9%. The variability of the pooled incidence of pulmonary embolism in individual studies could be explained by the differences in the disease severity and frequency of diagnostic imaging. Several mechanisms could explain the elevated incidence of pulmonary embolism in patients with COVID-19, including that these patients present with abnormally elevated levels of proinflammatory cytokines [5]. The elevated systemic inflammation could be related to endothelial injury by attachment of the virus to the angiotensin-2 receptor of the endothelial cells and viral replication, which could cause prothrombotic endothelial dysfunction [58, 59]. Moreover, the use of central venous catheters, mechanical ventilation, platelet activation, immobilization, and the other characteristics of COVID-19 therapies could play an important role in the prothrombotic state. In addition, these factors could explain the significant heterogeneity in the pooled incidence of pulmonary embolism in patients infected with COVID-19.

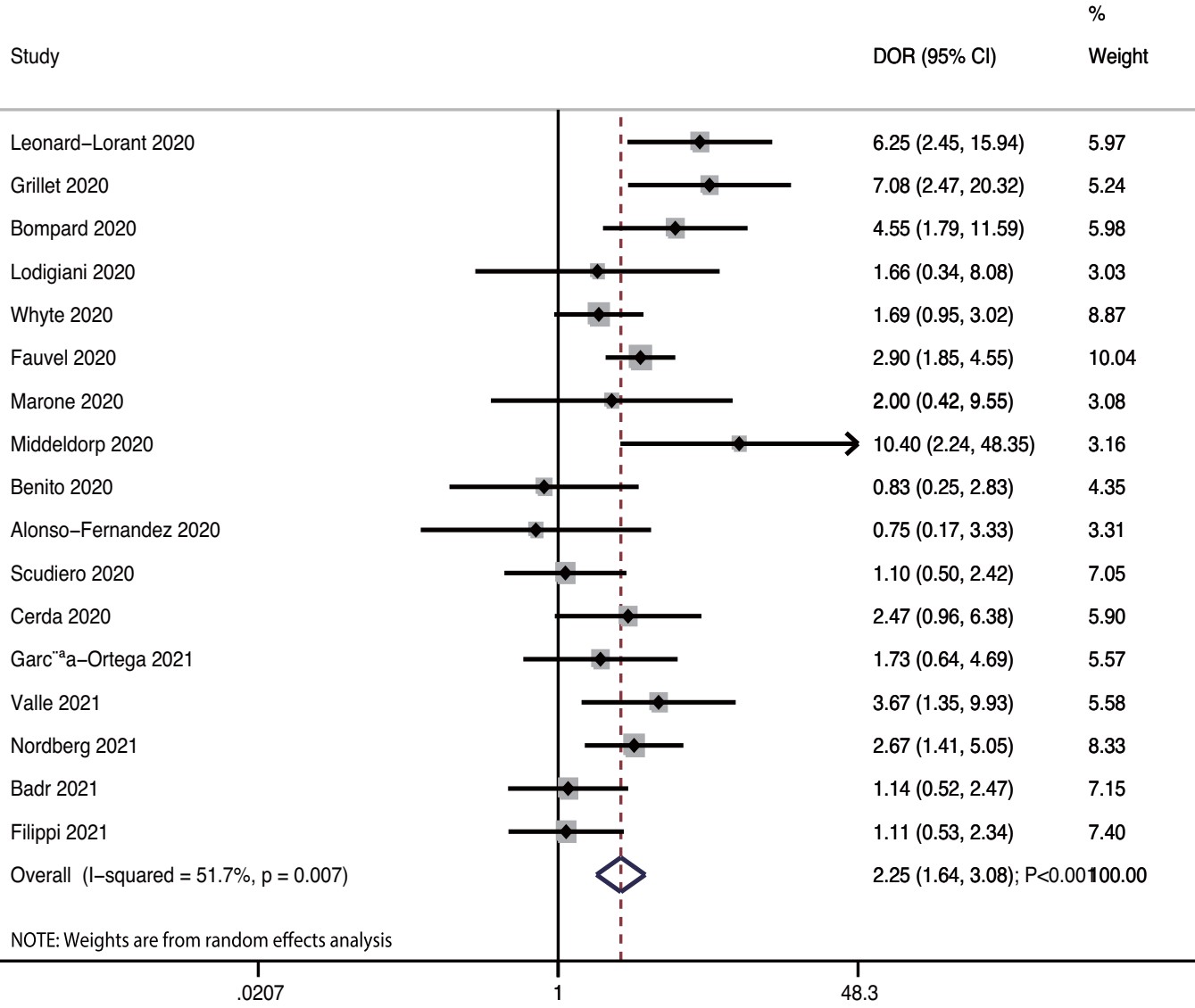

**Fig 5. Pooled DOR for pulmonary embolism on subsequent ICU admission.**

This study found that the pooled incidence of pulmonary embolism in patients with COVID-19 in the ICU was higher than that in non-ICU patients. Moreover, we noted that the predictive value of pulmonary embolism for ICU admission was statistically significant, although the predictive value of pulmonary embolism for ICU admission was mild. The higher incidence of pulmonary embolism in ICU patients could be explained by the severe procoagulant state in patients with COVID-19 presenting in a critical condition [60, 61]. However, we noted significant heterogeneity among the included studies in the relationship between pulmonary embolism and ICU admission, which could be explained by the variability in disease status, inflammatory status, and the diagnostic imaging modality used.

The limitations of this study should be acknowledged. This study was both a prospective and retrospective study. Thus, the results may have been affected by the uncontrolled biases from each study design. The diagnostic modality and frequency also differed amongst the studies, which could have affected the number of pulmonary embolisms diagnosed. In

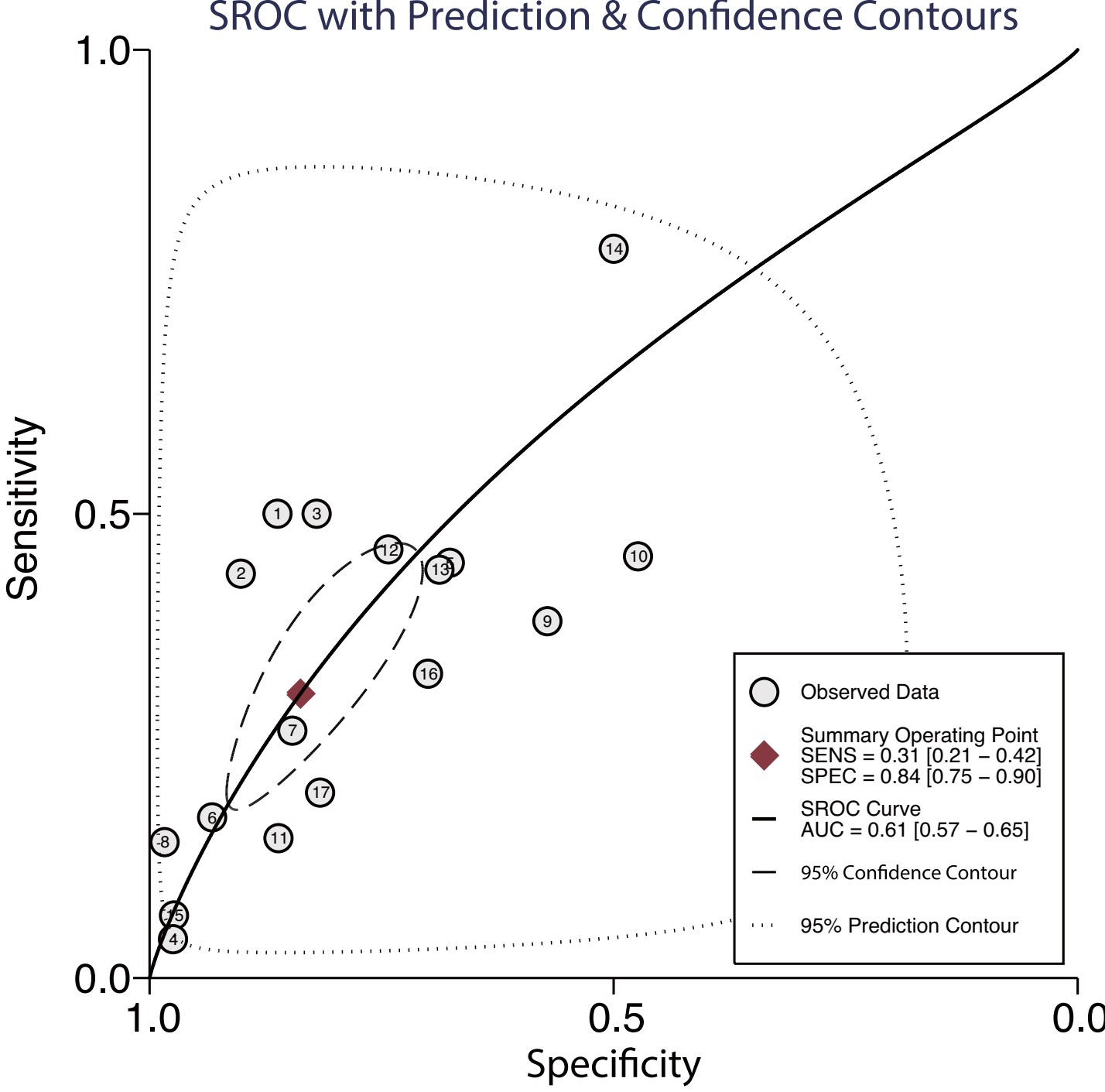

**Fig 6. AUC for the role of pulmonary embolism in intensive care unit (ICU) admission.**

addition, the significant heterogeneity was not further analyzed because the number of studies reporting the predictive value of pulmonary embolism for ICU admission was small. More-over, publication bias was inevitable because of the nature of the review and meta-analysis, which was based on the currently published articles. This study was not registered in

**Table 2. Subgroup analyses for the predictive value of pulmonary embolism for ICU admission.**

| Factors | Subgroup | DOR and 95%CI | *P* value | $I^2$ (%) | *P* value for $I^2$ | *P* value between subgroups |
|---|---|---|---|---|---|---|
| Study design | Prospective | 1.15 (0.58–2.29) | 0.688 | 0.0 | 0.543 | 0.046 |
| | Retrospective | 2.51 (1.79–3.51) | < 0.001 | 53.4 | 0.009 | |
| Mean age (years) | ≥ 65.0 | 1.88 (1.14–3.11) | 0.013 | 40.4 | 0.122 | 0.141 |
| | < 65.0 | 2.50 (1.66–3.74) | < 0.001 | 56.9 | 0.013 | |
| Male (%) | ≥ 70.0 | 3.09 (2.17–4.39) | < 0.001 | 0.0 | 0.523 | 0.030 |
| | < 70.0 | 1.81 (1.16–2.82) | 0.009 | 61.2 | 0.006 | |
| Study quality | High | 1.80 (1.14–2.84) | 0.012 | 54.8 | 0.039 | 0.118 |
| | Low | 2.71 (1.75–4.19) | < 0.001 | 48.3 | 0.043 | |

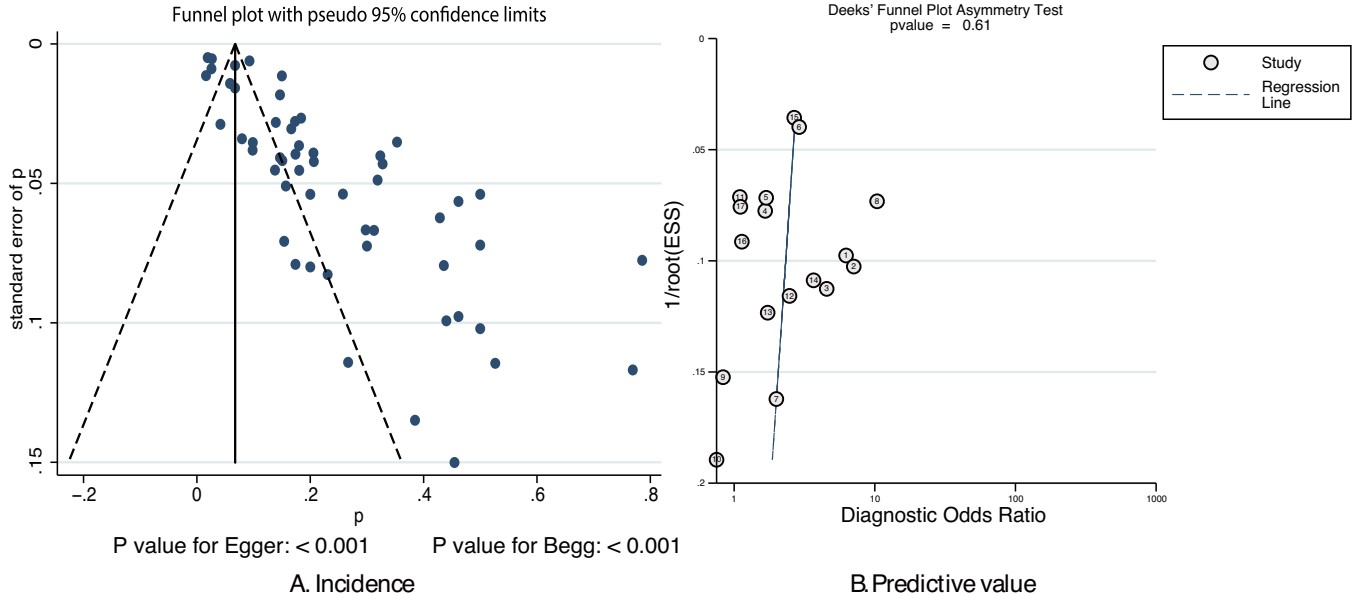

**Fig 7.** Funnel plot for the incidence of pulmonary embolism (A) and the predictive value of pulmonary embolism for ICU admission (B).

PROSPERO, and the transparency of this study could be affected; the analysis used pooled data, and the stratified analyses according to the characteristics of the patients were restricted.

## Conclusion

This study found that the pooled incidence of pulmonary embolism was 21% (95%CI: 18 −24%). Moreover, the incidence of pulmonary embolism in the ICU patients with COVID-19 was higher than that in the non-ICU patients with COVID-19. Moreover, the predictive value of pulmonary embolism for ICU admission was mild. Further research should be conducted to assess the role of pulmonary embolism in the prognosis of patients with COVID-19.

## Supporting information

**S1 Checklist. PRISMA checklist.**
(DOC)

## Author Contributions

**Conceptualization:** Xiaowei Gong.

**Data curation:** Xiaowei Gong.

**Formal analysis:** Xiaowei Gong, Boyun Yuan.

**Funding acquisition:** Yadong Yuan.

**Investigation:** Boyun Yuan.

**Methodology:** Xiaowei Gong, Boyun Yuan.

**Project administration:** Boyun Yuan, Yadong Yuan.

**Resources:** Xiaowei Gong, Boyun Yuan.

**Software:** Boyun Yuan.

**Supervision:** Boyun Yuan.

**Validation:** Yadong Yuan.

**Visualization:** Boyun Yuan, Yadong Yuan.

**Writing – original draft:** Xiaowei Gong.

**Writing – review & editing:** Yadong Yuan.

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
