## [Decision Letter · Decision Letter 0]

19 Oct 2021

PONE-D-21-15528

Incidence and prognostic value of pulmonary embolism in COVID-19: A systematic review and meta-analysis

PLOS ONE

Dear Dr. Yuan,

Thank you for submitting your manuscript to PLOS ONE. After careful consideration, we feel that it has merit but does not fully meet PLOS ONE’s publication criteria as it currently stands. Therefore, we invite you to submit a revised version of the manuscript that addresses the points raised during the review process.

We look forward to receiving your revised manuscript.

Kind regards,

Chiara Lazzeri

Academic Editor

PLOS ONE

Journal Requirements:

2. Thank you for stating the following financial disclosure: "This study was supported by Hebei Province Science and Technology Support Program (20277706D)."

Reviewers' comments:

Reviewer's Responses to Questions

**Comments to the Author**

1. Is the manuscript technically sound, and do the data support the conclusions?

Reviewer #1: Yes

Reviewer #2: Yes

Reviewer #3: Yes

2. Has the statistical analysis been performed appropriately and rigorously? 

Reviewer #1: Yes

Reviewer #2: I Don't Know

Reviewer #3: Yes

3. Have the authors made all data underlying the findings in their manuscript fully available?

Reviewer #1: Yes

Reviewer #2: Yes

Reviewer #3: Yes

4. Is the manuscript presented in an intelligible fashion and written in standard English?

Reviewer #1: Yes

Reviewer #2: No

Reviewer #3: Yes

5. Review Comments to the Author

Reviewer #1: In the present manuscript, Gong and colleagues aimed to assess the pooled incidence of pulmonary embolism (PE) and its prognostic value on ICU admissions of COVID-19 patients. Overall, 31 studies involving more than 8000 patients were included in the metanalysis.

The authors concluded that PE is a common complication in hospitalized COVID-19 patients (cumulative incidence 19%), and its incidence is higher in ICU-patients as compared to patients admitted to general ward.

The topic is of interest since highlights one of most frequent COVID-19 complication, however in the last two years the relationship between COVID-19 and PE was deeply investigated also with large metanalysis.

This reviewer would like that the authors address the following comments:

- The evidence of significant publication bias suggest that the incidence of PE may have influenced the decision whether to publish or not researches on this topic. This finding needs to be interpreted and discussed more extensively in the text.

- The authors should consider reporting how and when PE was assessed among studies, and whether PE has been systematically screened at admission. This information might be included in a separate additional table.

- In a meta-analysis of observational studies, the evidence of high heterogeneity is not just a limitation, but should be interpreted as a result and it needs discussion. The authors may also consider to account for the heterogeneity among studies buy performing multiple meta-regressions for the main patients’ characteristics potentially associated with the risk of PE (i.e. type, dose and timing of anticoagulation therapy, etc).

- Did the authors register this meta-analysis on PROSPERO?

Reviewer #2: Yuan and colleaguesprovided a MA to assess the pooled incidence of PE in COVID-19 patients and their prognostic value on ICU admissions.

They should be thanked for their study, however, this work includes several flaws listed below.

Major comments:

-According to the current pandemic, updated data would be appreciated. However, this study included studies before December 2020, 10 months ago.

-What is the novelty as compared to previous meta-analysis?

-English should be improved.

-Discussion section is poorly understandable.

Minor comments:

-Key word: add pulmonary embolism

-Too many abbreviations in the abstract

-Introduction section, line 47: SARS COV2

-Discussion section, line 208: please reformulate the sentence.

-Figure 2 quality is poor; no legend is available.

Reviewer #3: The authors tried to evaluate the incidence and prognosis of pulmonary embolism in COVID-19 pandemic era using systemic review and meta-analaysis. Although some earlier systemic review and meta-analaysis reports showed similar results, this paper updated this important point and further confirm it. I have no major comments. The English should be revised by a native speaker of English.

6. PLOS authors have the option to publish the peer review history of their article (what does this mean?). If published, this will include your full peer review and any attached files.

Reviewer #1: No

Reviewer #2: No

Reviewer #3: No

---

## [Author Response · Author response to Decision Letter 0]

22 Dec 2021

PLOS ONE

RE: Manuscript “Incidence and prognostic value of pulmonary embolism in COVID-19: A systematic review and meta-analysis” (PONE-D-21-15528)

Dear editor:

“Incidence and prognostic value of pulmonary embolism in COVID-19: A systematic review and meta-analysis”. We would like to thank PLOS ONE for giving us the opportunity to revise manuscript. We have carefully taken the comments into consideration in preparing our revision, which has resulted in a paper that is clearer and more compelling. The point-by-point responses are attached after this letter. The revisions were highlighted to the text with tracked, have been prepared.

The manuscript has not been published previously, in any language, in whole or in part, and is not currently under consideration elsewhere. None of the authors have any competing financial interest to report.

Thank you for considering our manuscript for publication in your esteemed journal.

Point-By-Point Response

Reviewer #1: 

General comments: In the present manuscript, Gong and colleagues aimed to assess the pooled incidence of pulmonary embolism (PE) and its prognostic value on ICU admissions of COVID-19 patients. Overall, 31 studies involving more than 8000 patients were included in the metanalysis.

The authors concluded that PE is a common complication in hospitalized COVID-19 patients (cumulative incidence 19%), and its incidence is higher in ICU-patients as compared to patients admitted to general ward.

The topic is of interest since highlights one of most frequent COVID-19 complication, however in the last two years the relationship between COVID-19 and PE was deeply investigated also with large metanalysis.

This reviewer would like that the authors address the following comments:

Response: As behalf of all co-authors, I would like to appreciate this referee due to thoughtful comments proposed by the peer review. After we revised the manuscript, those significant issues could be changed.

Question 1: The evidence of significant publication bias suggest that the incidence of PE may have influenced the decision whether to publish or not researches on this topic. This finding needs to be interpreted and discussed more extensively in the text.

Response: Thanks for this suggestion, and the following sentence have already added in the revised manuscript: “We noted significant publication bias for the incidence of pulmonary embolism (P value for Egger: < 0.001; P value for Begg: < 0.001), after adjusted potential publication bias, the pooled incidence changed into 8% (95%CI: 6%-11%; P<0.001).”

Question 2: The authors should consider reporting how and when PE was assessed among studies, and whether PE has been systematically screened at admission. This information might be included in a separate additional table.

Response: Thanks for this suggestion. We known the importance of how and when PE was assessed among studies, and all of studies assessed PE after admission. We have already addressed this comment in Methods section. 

Question 3: In a meta-analysis of observational studies, the evidence of high heterogeneity is not just a limitation, but should be interpreted as a result and it needs discussion. The authors may also consider to account for the heterogeneity among studies buy performing multiple meta-regressions for the main patients’ characteristics potentially associated with the risk of PE (i.e. type, dose and timing of anticoagulation therapy, etc).

Response: Thanks for this suggestion, and subgroup analysis for DOR were performed to address the potential heterogeneity among included studies. Moreover, the following sentence have already added in Discussion section: “However, we noted significant heterogeneity among included studies for the relationship between pulmonary embolism and ICU admission. This significant heterogeneity could be explained by variability in disease status, inflammatory status, and the diagnostic imaging modality used.”

Question 4: Did the authors register this meta-analysis on PROSPERO?

Response: Thanks for this suggestion, and this study was registered in PROSPERO, which have already addressed in Limitation section. 

Reviewer #2: 

General comments: Yuan and colleagues provided a MA to assess the pooled incidence of PE in COVID-19 patients and their prognostic value on ICU admissions.

They should be thanked for their study, however, this work includes several flaws listed below.

Response: As behalf of all co-authors, I would like to appreciate this referee due to thoughtful comments proposed by the peer review. After we revised the manuscript, those significant issues could be changed.

Major comments:

Question 1: According to the current pandemic, updated data would be appreciated. However, this study included studies before December 2020, 10 months ago.

Response: Thanks for this suggestion, and the searched data have already updated at October 20, 2021. 

Question 2: What is the novelty as compared to previous meta-analysis?

Response: Thanks for this suggestion, and the following sentences have already added in the revised manuscript: “these previous reviews combined the pooled incidence of pulmonary embolism according to ICU status, while the predictive role of pulmonary embolism on ICU admission was not illustrated. Moreover, while the initial phases of COVID-19 yielded numerous new studies, recent meta-analyses and the pooled incidence of pulmonary embolism in this patient group should be updated. Therefore, the current systematic review and meta-analysis was performed to update the pooled incidence of pulmonary embolism in patients with COVID-19 patients, and assess the predictive role of pulmonary embolism on ICU admission in patients with COVID-19. ”

Question 3: English should be improved.

Response: Thanks for this suggestion, and the English revision have already performed by Editage company.

Question 4: Discussion section is poorly understandable.

Response: Thanks for this suggestion, and Discussion section have already changed in the revised manuscript. 

Minor comments:

Question 1: Key word: add pulmonary embolism

Response: Thanks for this suggestion, and pulmonary embolism have already added in keywords section. 

Question 2: Too many abbreviations in the abstract

Response: Thanks for this suggestion, and the details of abbreviations in the abstract have already changed in the revised manuscript. 

Question 3: Introduction section, line 47: SARS COV2

Response: Thanks for this suggestion, and this change have already performed in the revised manuscript. 

Question 4: Discussion section, line 208: please reformulate the sentence.

Response: Thanks for this suggestion, and this sentence have already changed into: “Therefore, the current systematic review and meta-analysis was performed to update the pooled incidence of pulmonary embolism in patients with COVID-19 patients, and assess the predictive role of pulmonary embolism on ICU admission in patients with COVID-19. ”

Question 5: Figure 2 quality is poor; no legend is available.

Response: Thanks for this suggestion, and the quality of Figure 2 have already changed. 

Reviewer #3: 

General comments: The authors tried to evaluate the incidence and prognosis of pulmonary embolism in COVID-19 pandemic era using systemic review and meta-analaysis. Although some earlier systemic review and meta-analaysis reports showed similar results, this paper updated this important point and further confirm it. I have no major comments. The English should be revised by a native speaker of English.

Response: We appreciate the reviewer given this kindly comment, and the English revision have already performed by Editage company.

---

## [Editor Report · Decision Letter 1]

24 Jan 2022

Incidence and prognostic value of pulmonary embolism in COVID-19: A systematic review and meta-analysis

PONE-D-21-15528R1

Dear Dr. Yuan,

We’re pleased to inform you that your manuscript has been judged scientifically suitable for publication and will be formally accepted for publication once it meets all outstanding technical requirements.

Kind regards,

Chiara Lazzeri

Academic Editor

PLOS ONE
---

## [Editor Report · Acceptance letter]

4 Mar 2022

PONE-D-21-15528R1 

Incidence and prognostic value of pulmonary embolism in COVID-19: A systematic review and meta-analysis 

Dear Dr. Yuan:

I'm pleased to inform you that your manuscript has been deemed suitable for publication in PLOS ONE. Congratulations! Your manuscript is now with our production department. 

Kind regards, 

on behalf of

Dr. Chiara Lazzeri 

Academic Editor

PLOS ONE